# Learning to Fly Camera Drones by Watching Internet Videos

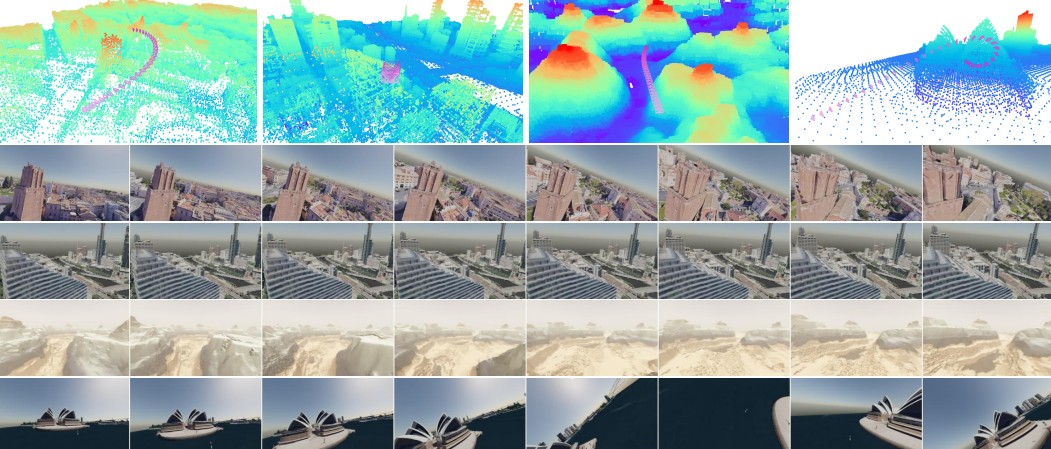

Figure 1: Examples of generated camera drone trajectories and their corresponding videos recorded from simulation engines. Learned by watching online videos, the proposed method can generate diverse trajectories with smooth movements, conditioned on drone types and captured images.

## Abstract

Camera drones offer unique perspectives and dynamic motions, yet automating their control for drone videography remains an open question. Unlike navigation or racing, there is no well-accepted reward function for human viewing experiences, making reinforcement learning approaches ill-suited. Therefore, we propose an imitation learning pipeline that learns from Internet videos by mimicking expert operations. In the absence of teleoperation data such as controller inputs and flight logs, we use reconstructed 3D camera poses to estimate camera drone trajectories. Importantly, to ensure data quality, we develop a scalable filtering scheme based on trajectory smoothness. After discarding more than three quarters of processed data, we produce 99k high-quality trajectories, making it the largest camera drone motion dataset. To evaluate this new task, we introduce an interactive evaluation environment with 38 natural scenes and 7 real city scans, and benchmark metrics at both the instance and dataset levels. As a minor contribution, we present a strong baseline named DVGFormer. Despite architectural simplicity, the proposed approach can reproduce complex cinematic behaviors such as obstacle-aware weaving, scenic reveals, and orbiting shots, verifying the effectiveness of the proposed imitation learning formulation. Data and code are available at link.

## 1 Introduction

Camera drones can record videos from controllable locations and angles, making them ideal testbeds for AI videography systems (Mademlis et al., 2019b). Prior research on automated drone controls demonstrates great success in navigation (Li et al., 2020), obstacle avoidance (McGuire et al., 2017), and speed racing (Kaufmann et al., 2023). However, unlike these tasks, videography lacks a well-defined reward function that reflects human viewing experience. In fact, heuristic scores cannot cover diverse scenarios (Huang et al., 2018), and subjective aesthetics are difficult to quantify (Mademlis et al., 2024; Azzarelli et al., 2024). Such reward functions are essential for optimization (Nägeli et al., 2017) and learning-based methods (Luo et al., 2019), and their absence makes it particularly challenging to automate movement control for drone videography.

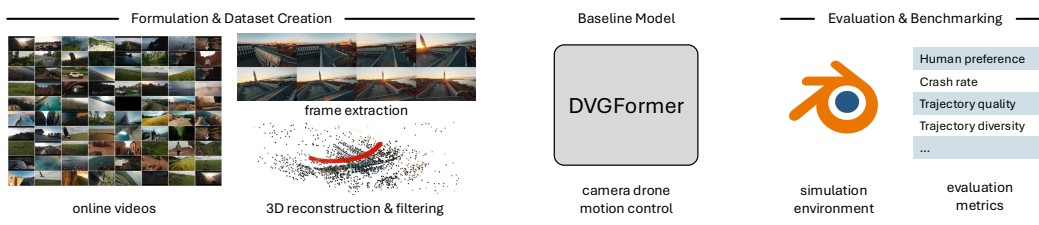

Figure 2: Contribution overview.

In this work, we propose an imitation learning approach that learns directly from online videos. Rather than relying on handcrafted heuristic rewards (Joubert et al., 2016; Huang et al., 2018; Mademlis et al., 2024), our system mimics expert operations in recorded drone footage. In terms of training data, one straightforward choice is to collect full teleoperation data (Brohan et al., 2022; Kim et al., 2024), such as controller joystick inputs and drone flight logs, but this is costly and typically lacks diversity. Instead, we leverage publicly available online videos, which are inexpensive to obtain and abundant. Specifically, we curate a large-scale real-world training set from YouTube drone footage. To prepare **drone trajectory ground truth**, we reconstruct the camera paths in 3D, and discard jittery camera trajectories using Kalman filter smoothness scores. After filtering out three-quarters of processed data, we produce 99,003 high-quality 3D drone trajectories in our DroneMotion-99k dataset.

Beyond the dataset, we introduce an interactive environment and a suite of metrics for evaluation. Prior studies on AI videography (Jiang et al., 2020; Courant et al., 2024) often rely on static and non-interactive evaluation protocols that compare against ground-truth motions from existing videos. In contrast, we argue that drone motions should directly influence their observations, and therefore customize Blender (Blender) to build an **interactive environment** that renders images conditioned on drone motions. We further propose a set of **metrics** that assess trajectory quality at both the instance and dataset levels, including human preference, crash rate, image-trajectory alignment, and diversity, establishing the first quantitative benchmark for drone videography.

As a minor contribution, we present DVGFormer, a baseline model that predicts camera trajectories conditioned on previously captured frames. While we do not aim for architectural novelty, our results demonstrate that with the right data and tokenization, even a standard Transformer (Radford et al., 2019) can generate sophisticated and diverse camera trajectories (Fig. 1). This validates the effectiveness of our imitation learning approach and highlights the central importance of data and problem framing. A summary of our contributions is provided in Fig. 2.

## 2 RELATED WORK

**AI cinematography and videography** often use *heuristics*, such as keeping the actors in frame and maintaining distance (He et al., 1996; Yu et al., 2022a;b; Azzarelli et al., 2024). Another option is *exemplar videos* (Jiang et al., 2020; 2021; Wang et al., 2023a), where the system transfers shot types from video examples to create new shots. *Textual prompts* for shot guidance recently received wide attention (Courant et al., 2024; Jiang et al., 2024; Li et al., 2024). However, these methods often overlook the interactive nature of control, and do not consider collision in their evaluation.

**Drone videography** is mostly heuristic-based (Joubert et al., 2016; Nägeli et al., 2017; Huang et al., 2018; Mademlis et al., 2019a; 2024). However, these heuristics-based rewards cannot well characterize the human viewing experience, making it difficult to train on simulation platforms (Jeon et al., 2020b;a; Pueyo et al., 2024). In an effort to learn from real data, Huang et al. (2019) learn optical flow as a surrogate of the camera pose. Ashtari et al. (2022) collect the DVCD18K dataset from online videos, but it lacks high-quality 3D control signal ground truth, which is essential in imitation learning. In comparison, this research aims to learn from a real-world training dataset with high-quality 3D camera trajectory, and test on a simulation-based interactive testbed.

**Camera movement conditioning in video generation** attracts a lot of attention recently (Zhang et al., 2024; Namekata et al., 2024; Kuang et al., 2024; He et al., 2024; Xu et al., 2024; Bahmani et al., 2024; Wang et al., 2023b). For example, CameraCtrl (He et al., 2024) designs a plugin module for the video generation framework. Note that these methods require *predefined* camera paths, while our goal is to *generate* camera trajectories.

Table 1: Comparison with existing camera trajectory datasets. The DroneMotion-99k dataset has the longest total duration and high quality 3D trajectory annotations under a low acquisition cost.

| | data source | shot type | camera motion | anno. quality | cost | #clips | duration |
|---|---|---|---|---|---|---|---|
| Huang et al. (2019) | **drone videos** | tracking | optical flow (Ilg et al., 2017) | low | **low** | 92 | 0.6h |
| DCM (Wang et al., 2024) | 3D animations | tracking | professional animator | **very high** | high | 108 | 3.2h |
| DVCD18K (Ashtari et al., 2022) | **drone videos** | **general** | SLAM (Mur-Artal & Tardós, 2017) | low | **low** | 18k | 44.3h |
| CCD (Jiang et al., 2024) | 3D animations | tracking | professional animator | **very high** | high | 25k | 41.7h |
| E.T. (Courant et al., 2024) | movies | tracking | human-centered SfM (Ye et al., 2023) | medium | **low** | **115k** | 120.0h |
| DroneMotion-99k (**ours**) | **drone videos** | **general** | COLMAP + filtering | **high** | **low** | 99k | **182.3h** |

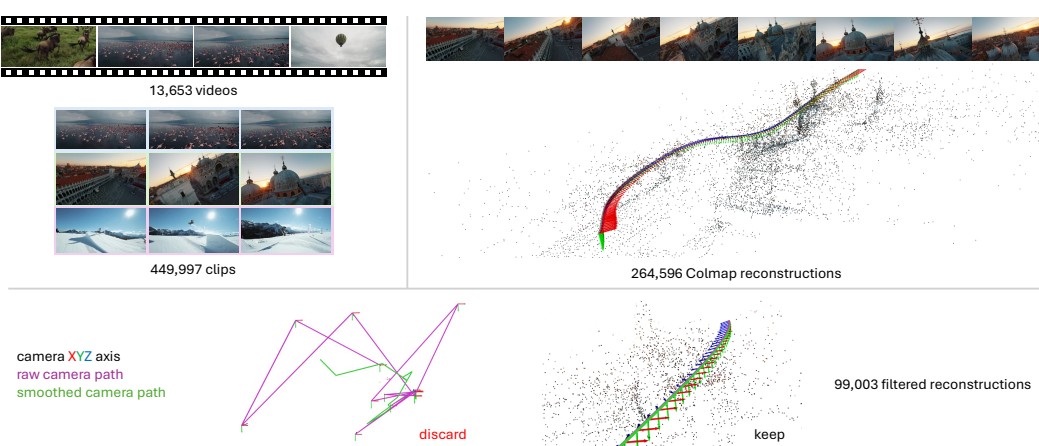

13,653 videos

449,997 clips

264,596 Colmap reconstructions

camera XYZ axis
raw camera path
smoothed camera path

99,003 filtered reconstructions

discard

keep

Figure 3: Data collection pipeline. **Top left**: We run shot change detection to split YouTube videos into clips of individual scenes. **Top right**: We then use COLMAP to reconstruct 3D scenes and recover camera poses from video frames. **Bottom**: Finally, we apply Kalman filter to 3D camera trajectories and discard low-quality reconstructions with jittery camera paths.

## 3 FORMULATION AND THE DRONEMOTION-99K DATASET

**Problem formulation.** We consider the camera pose $c$ and camera motion $a$ in COLMAP convention,

$$c = \{x, y, z, q_w, q_x, q_y, q_z\}, \tag{1}$$

$$a = \{v_x, v_y, v_z, \omega_x, \omega_y, \omega_z\} \tag{2}$$

where $x, y, z$ and $q_w, q_x, q_y, q_z$ denote camera location and rotation (quaternion). Linear velocity $v_x, v_y, v_z$ and angular velocity $\omega_x, \omega_y, \omega_z$ are expressed in the local coordinate frame defined by $c$.

Due to the lack of reward functions for human viewing experiences, we formulate the camera drone motion control problem as imitation learning. Specifically, at a time step $t$, given previous video frames $x_{0:t}$ and drone motions $a_{0:t}$, the motion control model learns to predict the next motion $a_{t+1}$, supervised by the ground truth action $a_{t+1,\text{gt}}$ from the dataset.

For imitation learning, it is essential to curate a large-scale dataset with *high-quality ground-truth annotations*. Rather than recording full teleoperations from human expert, we introduce a systematic approach in generating high-quality motion annotation from online videos at a low cost.

**Video pre-processing.** We first filter out the videos for harmful or weaponized usage of drones, and download 13k appropriate YouTube videos at 1080p resolution for a total duration of 1.5k hours. We split each video into clips of an individual scene (see Fig. 3 top left) with shot change detection (Castellano), ending up with ~450k video clips. We then filter out clips with conversations.

**3D reconstruction.** We use the Structure-from-Motion (SfM) method COLMAP (Schönberger & Frahm, 2016) to reconstruct the 3D camera path (Fig. 3 top right). To balance the computational cost and the reconstruction quality, we extract frames at 15 fps and 1080p, and apply affine-covariant feature detection and Domain-Size Pooling (Dong & Soatto, 2015) for SIFT (Lowe, 2004) features while limiting the feature point count to 512 per image. Using 4 threads per process, each COLMAP reconstruction worker takes roughly 200 seconds to finish on average. In total, the computation takes ~34k CPU hours, roughly three weeks on a 224-core CPU server, producing ~250k reconstructions. To address scale ambiguity, we normalize the reconstructed scene based on the average camera distance between neighboring frames, assuming a stable maximum speed of drones.

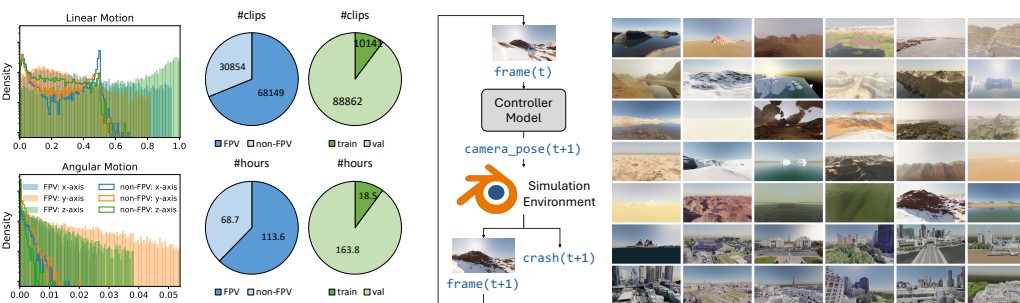

Figure 4: Data statistics.   Figure 5: Interactive evaluation and simulation scenes.

**Data filtering.** To filter out low-quality data, we identify 3D reconstructions whose camera locations from neighboring frames are drastically apart (Fig. 3 bottom), which is highly unlikely due to the continuous nature of camera movements. We adopt the Unscented Kalman Filter (Wan & Van Der Merwe, 2000) to produce a smoothed camera path, and compare the distance of the original camera locations to the smoothed camera path to identify reconstruction with reasonable movements. From ~1k human-labeled trajectories, we derive a threshold that best separates the correct and incorrect reconstructions (see Appendix A).

**Dataset statistics.** Overall, we collect ~99k samples for a total duration of ~180 hours. We compare our dataset with alternatives in Table 1. DroneMotion-99k covers a general topic and contains high-quality 3D camera paths that are more accurate than those generated without the filtering stage. Specifically, DroneMotion-99k covers two main types of drones, first-person-view (FPV) and non-FPV, with a roughly 2:1 ratio. As shown in Fig. 4, FPV drones often travel forward at a higher speed (larger $v_x$) with larger pitch ($\omega_y$), roll ($\omega_z$), and yaw ($\omega_x$) motions; while non-FPV drones have more stable shots (lower linear and angular speed). We split 10% of the full dataset into a validation set and use the remaining as the training set.

## 4 EVALUATION AND BENCHMARKING

In this research, we introduce an interactive evaluation platform and a set of metrics for drone videography. Unlike the static or offline setting in AI videography literature (Courant et al., 2024; Wang et al., 2024; Li et al., 2024; Jiang et al., 2021; 2024), our evaluation protocol does _NOT_ measure against "ground truth" motions in existing videos. Instead, it emphasizes interaction with and feedback from the environment. As shown in Fig. 5 left, the control model considers existing frames and predicts motion for the next frame, and the simulation environment renders the image and returns a binary crash indicator.

### 4.1 SIMULATION EVALUATION PLATFORM

To mimic video recording in the real world, we build an interactive simulation testbed by customizing the Blender engine. To balance the render quality and the speed, we choose a resolution of $225 \times 400$ and 64 rendering samples per frame.[1] By default, we generate 3D camera path for a 10-second duration. See Appendix B for more details.

For synthetic natural scenes, we use InfiniGen (Raistrick et al., 2023) to randomly generate 38 scenes from 10 templates `arctic`, `canyon`, `cliff`, `coast`, `desert`, `forest`, `mountain`, `plain`, `river`, `snowy_mountain`.

For real cities, we choose London, Paris, Rome, New York, Sydney, Melbourne, and Himeji. Regions of roughly 1km × 1km area are manually selected and the corresponding Google Earth 3D meshes are imported via the BLender Open StreetMap (BLOSM) toolkit (blo).

During evaluation, we run each scene 3 to 10 times with different initial camera poses and drone types for a total of 184 runs. Specifically, for FPV and non-FPV drones, we set their focal length to 12 and 24 mm, respectively, following typical specifications (dji, b;a).

---

[1]This low sample count introduces render noise/jitter; increasing samples alleviates this, but we prioritize speed over fidelity.

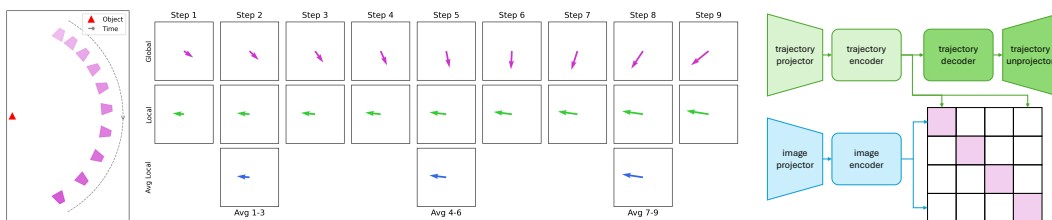

Figure 6: Kinetic feature. **Left**: Example of a 2D camera path. **Right**: Camera speed $\boldsymbol{v}$ in global frame (*top*), local frame (camera pointing up, *middle*), and averaged into $M = 3$ groups for $\boldsymbol{f}_{\text{kinetic}}$ (*bottom*).

Figure 7: CImTr feature trained from reconstruction and contrastive objectives.

## 4.2 EVALUATION METRICS

**Kinetic features.** To represent the camera trajectory over an arbitrary long period trajectory, we first consider the kinetic energy (Onuma et al., 2008) over different parts of the trajectory. Specifically, we first compute the linear speed for the generated camera trajectory $\boldsymbol{v} = \{v_x, v_y, v_z\}$ in the local coordinate system specified by the camera pose $\boldsymbol{c}$, where $v_x$ denotes the rightward speed, $v_y$ denotes the downward speed, and $v_z$ denotes the forward speed. Next, we downsample the sequence $t \in \{0, ..., T\}$ of an arbitrary length $T+1$ into $M$ groups, where each group $m \in \{1, ..., M\}$ corresponds to a sub-sequence $t \in \{T_{m-1}+1, ..., T_m\}$, where $T_m$ denotes the end of the $m$-th sub-sequence, $T_0 = 0, T_M = T$. We then average the speed vectors within that period $\boldsymbol{v}_m = \frac{1}{T_m - T_{m-1}} \sum_{T_{m-1}}^{T_m} \boldsymbol{v}_t$ (Fig. 6). Following this, we create the kinetic feature representation for the camera trajectory as,

$$\boldsymbol{f}_{\text{kinetic}} = \left\{ v_{m,x}^2, v_{m,y}^2, v_{m,z}^2 \right\} \in \mathbb{R}^{3 \times M}, \tag{3}$$

where $v_{m,x}, v_{m,y}, v_{m,z}$ denote the components of the average speed $\boldsymbol{v}_m$ for group $m$. Through monitoring the kinetic energy distribution over $M$ groups, this feature representation helps to measure the motion smoothness and the trajectory distribution. In practice, we set $M = 5$.

**CImTr features.** To measure alignment between images and trajectories, we introduce a Contrastive Image Trajectory (CImTr) feature between image feature $\boldsymbol{f}_{\text{CImTr}}^{\text{img}}$ and trajectory feature $\boldsymbol{f}_{\text{CImTr}}^{\text{traj}}$. Inspired by the text-to-human-motion retrieval method TMR (Petrovich et al., 2023), we combine the reconstruction objective $\mathcal{L}_{\text{recons}}$ and the contrastive objective $\mathcal{L}_{\text{contrast}}$ into an overall loss $\mathcal{L}_{\text{CImTr}} = \mathcal{L}_{\text{recons}} + \mathcal{L}_{\text{contrast}}$ for training the CImTr model. Specifically, we find that the image modality is less concise than the language modality, and the camera trajectory can also be redundant when using a higher frame rate (like the native data annotation at 15 fps). We thus introduce learnable projectors for the image and the trajectory as information bottlenecks, which greatly helps to reduce overfitting. A simple illustration can be found in Fig. 7. More details can be found in Appendix C.

**Quantitative metrics.** For *instance-level evaluation*, we consider the *image-trajectory alignment* between the initial frame and the predicted camera trajectory. Similar to CLIP-score (Hessel et al., 2021), we use the CImTr features to calculate this image-trajectory alignment score as $\texttt{CImTr-S} = 100 \times \max\left( cos\left( \boldsymbol{f}_{\text{CImTr}}^{\text{img}}, \boldsymbol{f}_{\text{CImTr}}^{\text{traj}} \right), 0 \right)$. We also evaluate *obstacle avoidance* with the crash rate using a distance clearance of 0.5 meters. Finally, we report *human preference* with a reference model.

For *dataset-level evaluation*, we measure trajectory quality using Frechet Inception Distance (FID) (Heusel et al., 2017) on kinetic feature $\boldsymbol{f}_{\text{kinetic}}$ and CImTr feature $\boldsymbol{f}_{\text{CImTr}}^{\text{traj}}$. FID measures the distribution distance between generated camera trajectories and validation set trajectories. We also measure *trajectory diversity* following (Li et al., 2021; Siyao et al., 2022; Wang et al., 2024) and report the average pair-wise feature distance $\texttt{div}_{\text{CImTr}} = \mathbb{E}_{i,j} \| \boldsymbol{f}_{\text{CImTr},i}^{\text{traj}} - \boldsymbol{f}_{\text{CImTr},j}^{\text{traj}} \|_2$.

## 4.3 DISCUSSION

**Comparison with existing metrics.** Existing kinetic-based human motion (Onuma et al., 2008) or camera motion (Wang et al., 2024) metrics split variable-length sequences into fixed-length chunks, and represent one sequence with multiple chunks. In comparison, our group-based approach can represent one sequence as one feature vector, and reflect the motion smoothness across the sequence.

**Measuring image-trajectory alignment with `CImTr-S`.** Trained in a contrastive manner, CImTr feature distance can reflect how well trajectories align with images, and thus indicates the shot

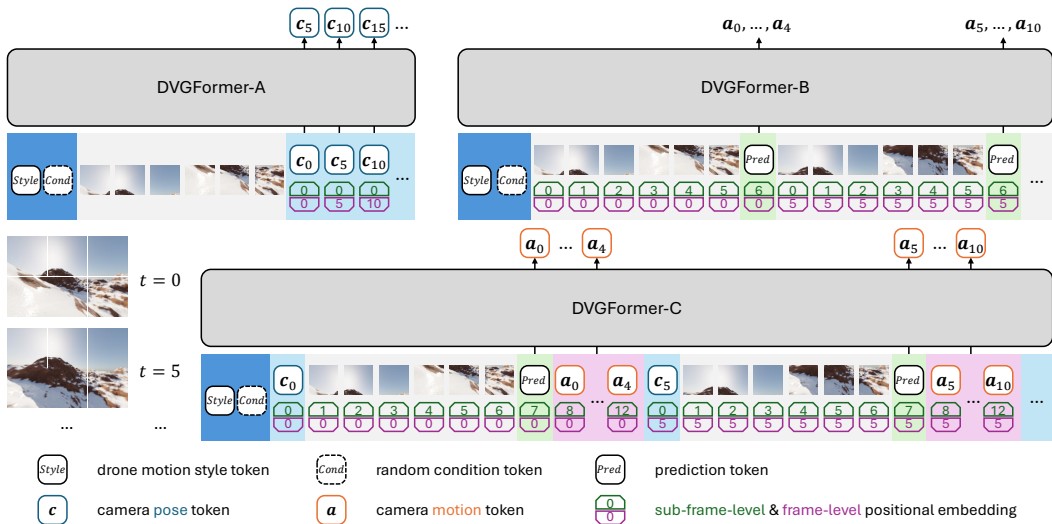

Figure 8: DVGFormer predicts the camera trajectory based on (A) the initial image and previous camera poses, (B) all previous images, and (C) previous camera poses, images, and motions.

planning quality. As a sanity check, we randomly swap the image-trajectory pairs on the DroneMotion-99k validation set, and find that this mismatch drops CImTr score from 38.859 to 4.882.

**Trajectory features using kinetic and CImTr representations.** $f_{\text{kinetic}}$ focuses on the linear motion of the trajectory and offers better explainability and alignment with existing literature (Li et al., 2021; Siyao et al., 2022; Wang et al., 2024), while $f_{\text{CImTr}}^{\text{traj}}$ provides an overall evaluation of both linear and angular velocities using the learned representation.

## 5 DVGFORMER MODEL

We consider a trajectory of camera poses $c_{0:T}$ (Eq. 1) and their corresponding images $x_{0:T}$. Our goal is to predict the future camera poses $c_{t:T}, t \in \{1, ..., T\}$ based on $c_{0:t-1}$ and $x_{0:t-1}$, through either estimating the camera pose $c$ directly or the camera motion $a$ (Eq. 2).

We build our Drone Videography Transformer (DVGFormer) models with a standard autoregressive Transformer (Vaswani et al., 2017) architecture. For tokenization, first, we have two tokens for the *sequence-level* conditioning. A <Style> token encodes the drone type as either FPV or non-FPV drones, and a <Cond> token is sampled from a random Gaussian noise to further introduce randomness to the camera path prediction. We adopt DINOv2 (Oquab et al., 2023) to tokenize RGB image $x$, and MLPs to tokenize camera pose $c$ or motion $a$. For efficiency, we select a lower frame rate for the camera pose and image. In practice, we set the image frame rate (3 fps) to 5 times lower than the camera motion (15 fps). An overview of the DVGFormer model family can be found in Fig. 8 (refer to Appendix D for details).

**Bi-level positional embeddings.** Sequential orders at both frame and sub-frame levels are very important. Therefore, we introduce a bi-level design with interleaved frame-level and sub-frame-level positional embeddings. Compared to using different tokens for different locations in the sequence (Janner et al., 2021; Brohan et al., 2022; 2023), this bi-level design introduces fewer parameters and is less susceptible to overfitting. Compared to the frame-level-only design (Chen et al., 2021), this approach can differentiate different tokens within one time step, making it possible to contrast the same image patch over time for motion extraction.

To evaluate different designs for this newly introduced task, we consider three baselines:

**DVGFormer-A.** We condition only on the initial frame $x_0$ to mimic the offline configuration in previous works (Courant et al., 2024; Wang et al., 2024; Li et al., 2024; Jiang et al., 2021; 2024). We tokenize both the initial frame and the existing camera poses to represent the camera trajectory.

Table 2: **Quantitative performance**. The DVGFormer models (B and C) establish a strong baseline for this new task. Value differences are statistically significant ($p < 0.01$) over 5 runs. * denotes our implementation. "validation statistics" refers to the subset of DroneMotion-99k dataset.

| | input | | | prediction | instance-level | | | dataset-level | | |
|---|---|---|---|---|---|---|---|---|---|---|
| | image | pose | motion | | CImTr-S↑ | crash↓ (%) | prefer.↑ (%) | FID_kinetic↓ | FID_ClmTr↓ | div_ClmTr↑ |
| validation statistics | - | - | - | - | 38.859 | - | - | reference | reference | 1.410 |
| DIRECTOR* | N/A | - | - | pose | 5.289 | 15.217 | 15.3±3.7 | 44.283 | 1.196 | 0.952 |
| GenDoP* | init. img (RGBD) | ✓ | ✗ | pose | 6.926 | 9.239 | 21.7±4.3 | 35.780 | 1.021 | 1.004 |
| DVGFormer-A | init. img | ✓ | ✗ | pose | 8.896 | 16.304 | 23.3±4.1 | 35.111 | 0.872 | 1.179 |
| DVGFormer-B | 10 seconds | ✗ | ✗ | motion | 8.001 | 8.696 | 56.3±6.7 | 3.515 | 0.700 | 1.209 |
| DVGFormer-C | 10 seconds | ✓ | ✓ | motion | 9.277 | 3.804 | reference | 21.161 | 0.780 | 1.201 |
| Variant 1: prediction | 10 seconds | ✗ | ✗ | pose | 6.882 | 26.087 | - | 27.939 | 1.126 | 0.904 |
| Variant 2: prediction | 10 seconds | ✓ | ✗ | pose | 9.496 | 13.261 | - | 27.058 | 0.793 | 1.186 |
| Variant 3: tokenization | 10 seconds | ✗ | ✓ | motion | 6.826 | 12.500 | - | 18.101 | 0.736 | 1.220 |
| Variant 4: tokenization | 10 seconds | ✓ | ✗ | motion | 10.080 | 7.609 | - | 30.489 | 0.884 | 1.139 |
| Variant 5: image input | 2 seconds | ✗ | ✗ | motion | 6.560 | 14.130 | - | 18.667 | 0.733 | 1.145 |
| Variant 6: image input | 10 seconds (RGBD) | ✗ | ✗ | motion | 6.709 | 7.065 | - | 7.692 | 0.746 | 1.184 |

Following existing works, we also directly predict the camera pose $c$ of future frames instead of the camera motion $a$.

**DVGFormer-B.** We change from the offline configuration in DVGFormer-A to an interactive configuration, where the camera trajectory prediction is conditioned on previous frames $x_{0:t}$. We only tokenize the stream of images in this setting, making it comparable to an online video understanding model. In addition, we predict the camera motion $a$ from the prediction token <Pred>, which has a more stabilized distribution than the camera pose $c$. Motion prediction also supports a higher frame rate than pose prediction, which is one-to-one matched with image frames. In practice, we set the frame rate at 3 fps for images and camera poses, and 15 fps for motions.

**DVGFormer-C.** We further expand on the interactive design of DVGFormer-B and use previous camera poses $c_{0:t}$, images $x_{0:t}$, and motions $a_{0:t}$ as conditions. We discard the one-shot motion prediction from the <Pred> token and choose an iterative next token prediction approach.

**Training details.** For image-only augmentation, we apply the same random scaling and color jittering to the entire video. For trajectory-only augmentation, we apply random temporal clip to minimize the dependency on the starting frames. For image-and-trajectory augmentation, we apply random 3D horizontal flip, which flips both 2D images and the 3D camera pose and motion.

We use standard sequence-level loss to train DVGFormer, depending on the prediction, we have,

$$\mathcal{L} = \frac{1}{T} \sum_{t=1}^{T} \|c_{t,\text{pred}} - c_{t,\text{gt}}\|_1 \ \text{ or } \ \|a_{t,\text{pred}} - a_{t,\text{gt}}\|_1, \tag{4}$$

where $\| \cdot \|_1$ denotes the absolute distance to supervise the camera pose or motion predictions.

## 6 EXPERIMENTS

### 6.1 IMPLEMENTATION

We represent camera poses in global coordinates specified by the first frame, and camera motions in the local coordinates specified by the current frame. We use GPT-2 (Radford et al., 2019) for the auto-regressive transformer with 12 layers and a total of 40M parameters. We use the ViT-Small (Dosovitskiy et al., 2021) version of DINOv2 (Oquab et al., 2023), which is kept fixed during training. Experiments on two NVIDIA RTX-3090 GPUs take 8 hours for 10 epochs with a batch size of 32 per GPU and 4 gradient accumulation steps.

### 6.2 QUANTITATIVE RESULTS

In Table 2, we find that the offline option, **DVGFormer-A**, performs the worst in the DVGFormer family, highlighting the need for interactive inference. **DVGFormer-B** and **DVGFormer-C** establish strong performance, with the former prioritizing the trajectory quality (lowest FID values) and diversity (second best div), and the latter prioritizing the alignment between shot plans and image content (third best CImTr-S) and collision rates (lowest).

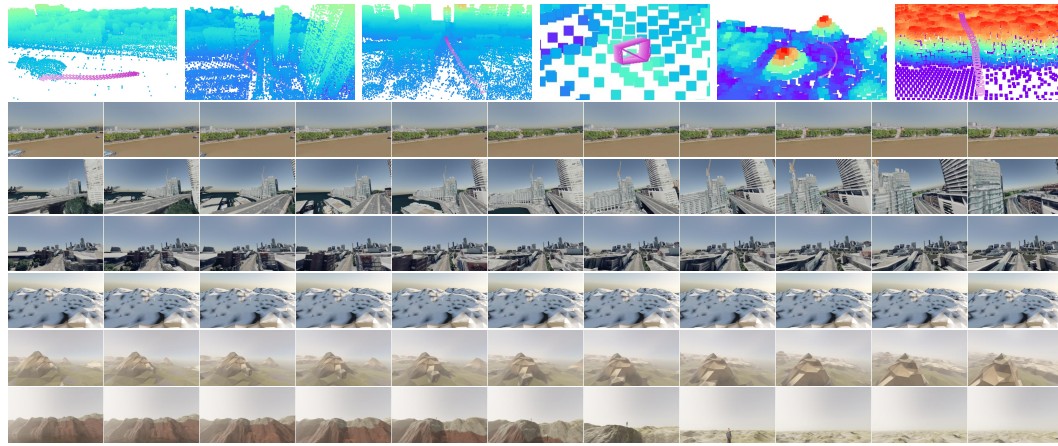

Figure 9: Result visualization. DVGFormer predicts camera trajectories (**top**), and the interactive environment renders them into videos (**bottom**). Generated camera motions include panning sideways, navigating through buildings, following the road, trucking backwards for a stable shot, orbiting mountains for a scenic reveal, or flying past the human actor at high speed.

We re-implement two models for comparison, *i.e.*, DIRECTOR (Courant et al., 2024), a diffusion-based trajectory generation model using human actor motion (unavailable) and text guidance, and GenDoP (Zhang et al., 2025), a concurrent auto-regressive pipeline that conditions on text prompts and initial image. We provide general text prompts "a video of an FPV drone" or "a video of a non-FPV drone". Due to the lack of image conditioning, camera trajectories from DIRECTOR (Courant et al., 2024) show poor alignment with images and a high crash rate. The monocular depth estimation (Yang et al., 2024) in GenDoP (Zhang et al., 2025) helps to lower the crash rate, but it also limits the trajectory alignment (lower `CImTr-S`) and the trajectory quality overall (worse `FIDs` and `div`).

The above quantitative results are also supported by **user preference**, where five human users are given 184 videos to rate their preference for each comparison. It is shown that for human viewers, motion quality (`FIDs` and `div`) might outweigh image-trajectory alignment (`CImTr-S`), but overall, the proposed metrics can well reflect the human preference. See Appendix E for more details.

For **variant studies**, in terms of *prediction type*, we find that directly swapping the prediction target from motion $a$ to camera pose $c$ significantly worsens the trajectory quality, diversity, and crash rate (Variant 1 vs. DVGFormer-B). Adding a pose token (Variant 2) helps with these issues and increases the image-trajectory alignment. However, it is not as competitive as DVGFormer-C overall.

For *tokenization*, Variant 3 adds motion tokens to DVGFormer-B and adopts an iterative prediction approach, which gives it slightly improved diversity at the cost of image-trajectory alignment. Variant 4 adds pose tokens to DVGFormer-B, which greatly improves the image-trajectory alignment at the cost of trajectory quality and diversity. Their combination, DVGFormer-C, achieves a good tradeoff between image-trajectory alignment and camera operation quality.

For *image input*, Variant 5 shortens the context window from 10 seconds in DVGFormer-B to 2 seconds, resulting in worse image-trajectory alignment and crash rate. Variant 6 adds depth estimation to DVGFormer-B. While slightly lowering the crash rate, this also hurts the shot planning capabilities.

## 6.3 QUALITATIVE RESULTS

We show videos generated by the proposed method (DVGFormer-B) in Fig. 1 and Fig. 9. Without any human guidance, the AI videography model generates diverse camera trajectories well-suited for each scenario, and can smoothly execute the camera movements.

**Controllable generation.** We can also control the generated camera path via `<Style>` and `<Cond>` tokens, as shown in Fig. 10. On the one hand, different `<Style>` settings affect both the camera focal lengths in the simulation engine and the predicted camera motion from DVGFormer. On the other hand, the <Cond> token (random noise) only affects the generated camera trajectories.

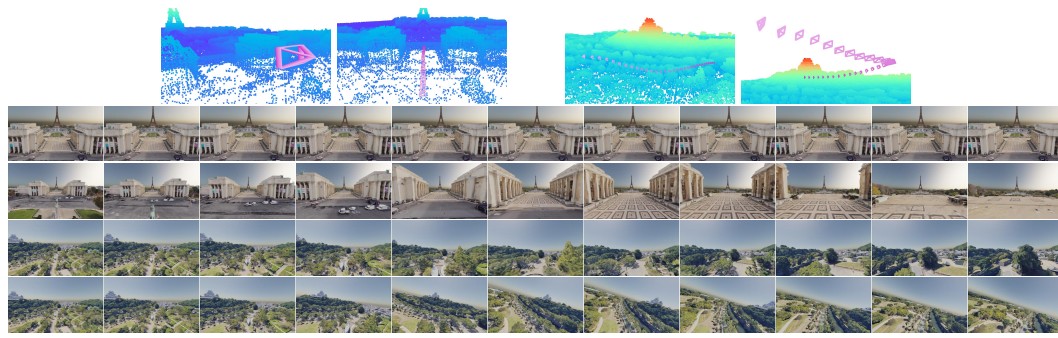

Figure 10: Generated camera trajectories from the same initialization (leftmost frame). **Top**: `<Style>` token specifies the drone being FPV or non-FPV, affecting camera focal length and drone motion style. **Bottom**: `<Cond>` token impacts the predicted camera path.

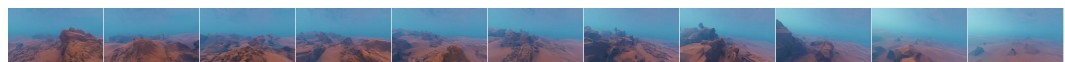

Figure 11: Generalization to unseen underwater scenarios.

**Generalization.** For unseen environments like underwater, the proposed DVGFormer can still translate its learned experience in drone videography to produce videos (Fig. 11).

## 7 LIMITATION AND FUTURE DIRECTIONS

**Scale ambiguity in monocular SfM** is a known issue. We experimented with metric depth estimation (Yin et al., 2023) to address this, but it did not yield reliable results in our setting. We thus normalize the scale in both the training data and the evaluation metric computation to minimize its influence.

**3D perception.** We currently use RGB images only in DVGFormer models. In our variant study, we find that monocular depth estimation results can help lower the crash rate. However, they might also bias the system and lead to lower trajectory diversity. Another point to note is that the monocular depth estimation results might not be accurate and consistent across the entire video sequence. Future works are encouraged to investigate sequence-level 3D perception methods like SLAM and further incorporate 3D point clouds into the conditioning.

**Implementation on drone hardware**. For piloting a drone in the real world, the current approach still has an undesirable crash rate. Plus, the predicted trajectories are a form of high-level control, and would need proper low-level controls on the drone motors. We expect future work to combine collision avoidance and low-level control into the overall optimization target.

**Language instructions** allow for granular controls, and we plan to investigate this in future works.

**Computational efficiency.** It takes 1.71 seconds (DVGFormer-A) to 1.93 seconds (DVGFormer-C) to predict the camera motion of a 10-second-long sequence on one RTX3090 GPU, making it real-time.

**Broader impact.** The task introduced, camera drone movement control, focuses on building AI assistants for content creation. For application on drone hardware, further safeguards can be considered to minimize potential collisions.

## 8 CONCLUSION

In this work, we investigate how to automate motion controls for camera drones in videography tasks, and introduce an imitation learning pipeline that learns from Internet videos. To this end, we (i) curate DroneMotion-99k, a large-scale dataset of real-world trajectories; (ii) establish an interactive benchmark with quantitative metrics for evaluating videography controls; and (iii) build DVGFormer, a strong baseline method. Results from DVGFormer demonstrate complex and diverse trajectories with smooth camera motions, validating the effectiveness of our imitation learning approach. We believe this work opens the door to scalable, reward-free learning of camera drone control policies, and ultimately to AI assistants that can autonomously film dynamic stories.

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

**Dataset Card:  DroneMotion-99k**

**Resource summary**

|         | #Clips | #Videos | Duration (hours) |
|---------|--------|---------|------------------|
| FPV     | 68,149 | 6,566   | 113.6            |
| Non-FPV | 30,854 | 2,969   | 68.7             |
| **Total** | **99,003** | **9,535** | **182.3**    |

**Split**: 90% train | 10% val (by *video*).
**Modality**: 3D camera trajectories only; RGB frames obtainable via the provided YouTube download scripts.
**Coordinate system**: COLMAP ($x$ rightward, $y$ downward, $z$ forward).
**Tasks**: camera movement generation, view-synthesis, robotics path planning, cinematic drone control.
**License**: CC-BY-NC-SA 4.0 for trajectory data. Original videos remain under their respective YouTube licenses.
**Collection**: 9,535 public YouTube videos automatically segmented into 99,003 clips; COLMAP used to reconstruct trajectories; manual filtering removes failure cases.
**Maintenance**: contact authors via GitHub issues. Takedown requests honored.
**Known limitations**: no absolute scale; users must comply with YouTube Terms of Service.

Figure 12: Dataset card summarizing the key properties of the DroneMotion-99k dataset.

Table 3: Total video duration (hours) at different stages of the data processing pipeline.

| raw videos | w/o portrait | clips w/o dialog | after 3D reconstruction | after filtering |
|------------|--------------|------------------|-------------------------|-----------------|
| 1,485      | 1,465        | 867              | 493                     | 182             |

## A  DATA COLLECTION DETAILS

**Overview**. A short data card is provided in Fig. 12 We compare the total video duration at each stage of the data processing pipeline in Table 3.

**Scraping videos.** We collect the dataset from YouTube videos. We search videos with keywords including "cinematic drone videos", "cinematic drone footage", "cinematic drone footage 4k", "cinematic fpv footage 4k", *etc.*, where FPV stands for first-person-view, a specific shot type that features drastic perspective changes to provide stimulating visual effects.

After searching YouTube videos, we first filter out ones with sensitive information including weaponized or harmful usage, for which we consider a blacklist of words including "weapon", "soldier", "attack", "strike", "military", "surveillance", "horror", *etc.*.

Next, we download the videos with the yt-dlp (ytd) package. We collected a total of 13,653 videos for a total duration of 1,485 hours.

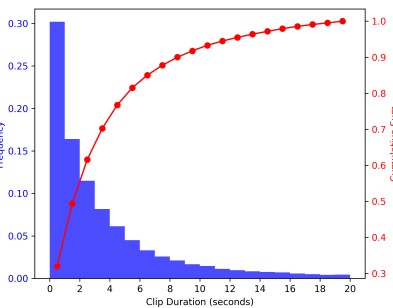

Figure 13: Clip duration distribution. 86.6% of the clips produced by shot change detection algorithm (Castellano) are 10 seconds or shorter. 69.4% of the clips are 1 second or longer.

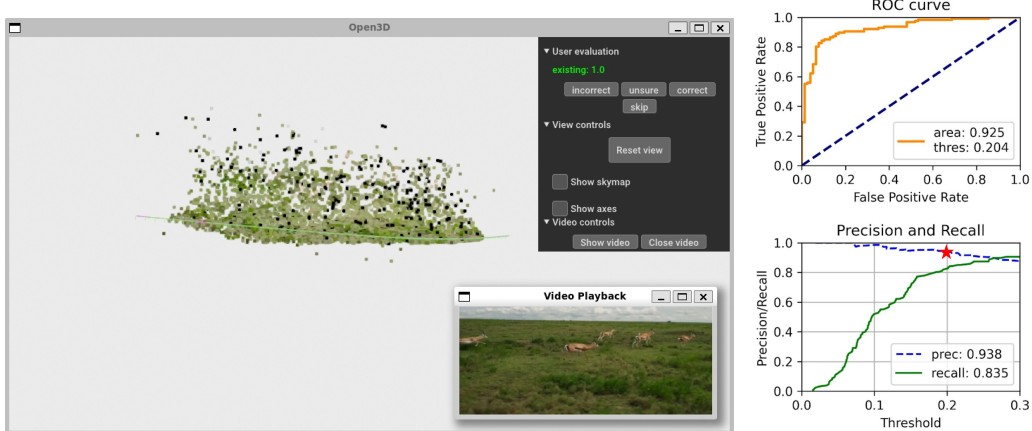

Figure 14: Threshold selection for identifying low-quality 3D reconstructions with unreasonable camera movements between consecutive frames. **Left**: We label the correctness of ~1k COLMAP reconstructions via our interactive 3D annotation tool by reviewing the reconstruction result and the original video clip side-by-side. **Right**: We gather statistics (ROC curve, precision, and recall) on the distance of camera locations to the smoothed camera path from Kalman filter, and select a threshold (red star) that best separates correct and incorrect reconstructions.

**Video data filtering and clip generation.** We only preserve those in the landscape mode and drop the portrait mode videos, since the natural sensor arrangement is the landscape mode. While 2,820 out of 13,653 videos are in the portrait mode, their total duration is much shorter in comparison, only 19.9 hours, which constitutes 1.3% of the total duration of 1,485 hours. This makes the following procedures much easier because the remaining videos are all of similar aspect ratios.

We then run PySceneDetect (Castellano) to detect the shot changes in videos, which produces a total of 642,806 individual clips. We show their duration distribution in Fig. 13. We filter out 30.6% of all clips whose duration is shorter than 1 second since they are too short and can be very difficult for the reconstruction. These sub-one-second fragments only make up for 2.1% of the total video duration. For the 13.4% of all clips whose duration is longer than 10 seconds, we break them down into partial clips with a maximum duration of 10 seconds.

We filter out clips with dialogue using the automatically generated or uploaded closed captions in YouTube videos. Apart from a few words in a whitelist, *e.g.*, "[music]", "[silence]", "[background noise]", "[pause]", "[sound effect]", the closed caption often suggests that the scene is unrelated to drone videos. After this step, there are 449,997 clips remaining for a total duration of 867 hours.

**3D reconstruction.** We extract frames from videos at a resolution of 1080p and a frame rate of 15 fps. In COLMAP (Schönberger & Frahm, 2016), we consider 512 feature points and extract SIFT (Lowe, 2004) features with `estimate_affine_shape=True` and `domain_size_pooling=True`. We also ignore the focal length changes in videos and set `single_camera=True` since we only focus on the change of camera location and direction in this work. We use `guided_matching=True` and the `sequential_matcher` for the frames extracted from videos. We run up to two sparse reconstruction steps, and when the first sparse reconstruction does not return any result, we extend some of the requirements and run a second round. This produces 264,596 reconstructions.

**Reconstruction filtering.** Reconstructions with a duration smaller than 15 frames or 1 second are discarded. We normalize the data according to the average difference in locations (speed). We discard data whose maximum camera speed exceeds 3 times the average camera speed.

We consider the reconstructions whose camera locations from neighboring frames are drastically apart as low-quality data. We design an automatic process for identifying and discarding those low-quality reconstructions. Specifically, we use Kalman filter to estimate a smoothed version of the camera trajectory, which then discloses how different it is compared to the original camera trajectory. If these two trajectories are drastically different, we then automatically discard the 3D reconstruction.

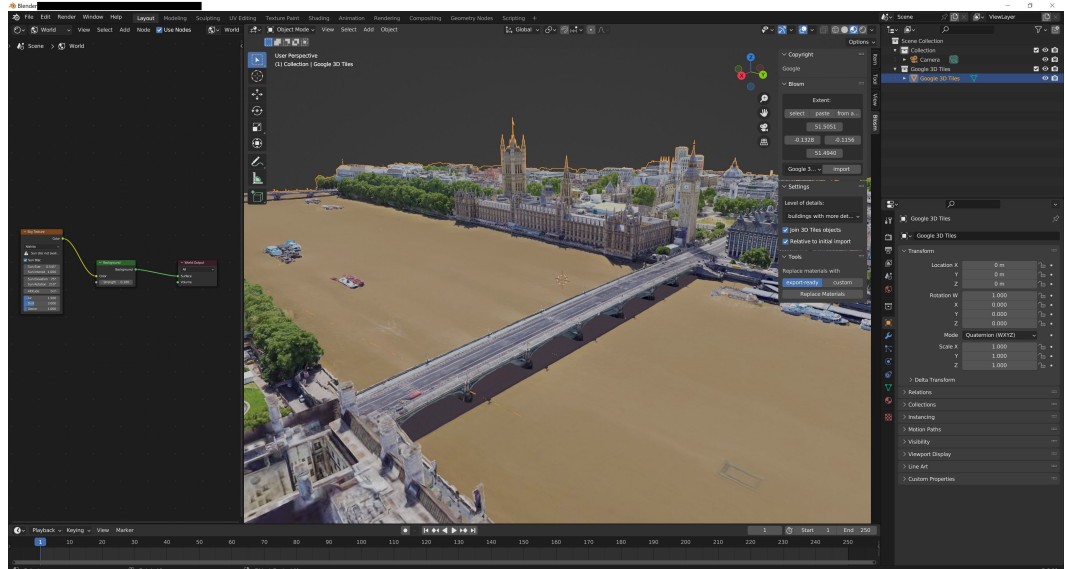

Figure 15: Simulation evaluation using Blender.

During implementation, we choose Unscented Kalman Filter (UKF) (Wan & Van Der Merwe, 2000) in FilterPy (fil). We consider a representation with 13 dimensions by combining the 7-dimensional translation vector and rotation quaternion in camera pose $c$ and the 6-dimensional speed and angular speed in camera motion $a$. We set $\alpha = 0.1$, $\beta = 2$, and $\kappa = 10$ for the hyperparameters in UKF.

Based on 1k labeled data from our interactive labeling tool, we select the optimal threshold for the difference between the original and the smoothed camera trajectory at 0.2 (see Fig 14). We end up with 99,003 camera trajectories with a total duration of 182 hours.

## B  SIMULATION TESTBED DETAILS

For synthetic natural scene in simulation experiments, we use InfiniGen (Raistrick et al., 2023) to randomly generate 38 scenes from 10 pre-defined natural scene types `arctic`, `canyon`, `cliff`, `coast`, `desert`, `forest`, `mountain`, `plain`, `river`, `snowy_mountain`. We use the `simple`, `no_assets`, and `no_creatures` settings to ensure the generated scene is not too complex. For generalization ability study in Fig 11, we use the `under_water` setting in InfiniGen. For scenes with human actor, we import free assets downloaded from the SketchFab website (ske).

For real cities in simulation experiments, we choose London, Paris, Rome, New York, Sydney, Melbourne, and Himeji. Regions of roughly 1km × 1km area are manually selected and the corresponding Google Earth 3D meshes are imported via the BLender Open StreetMap (BLOSM) toolkit (blo). We show a screenshot of the 3D city scan of London in Blender in Fig. 15.

During inference, we render the scenes with $225 \times 400$ resolution and 64 samples with a camera sensor width of 36mm. The lower resolution and number of samples in Blender (Blender) helps to increase the rendering speed. With that said, we can select an arbitrarily high resolution with higher sample quality during the final rendering, since we do not modify any 3D assets or 2D pixels and the videos are faithful depiction of the *existing* scene.

## C  CIMTR DETAILS

For the CImTr network, as shown in Fig. 7, we adopt a 4-layer transformer encoder architecture for both the image branch and the trajectory branch. We also adopt a 4-layer transformer decoder for the reconstruction objective for the trajectory branch. All CImTr transformers have 4 heads and a

Table 4: Retrieval performance on DroneMotion-99k validation set with 10141 samples. The top-$k$ recall (R@$k$) and mean Average Precision (mAP) are reported in percentages (%). We also report the median Rank (medR).

| | image to trajectory | | | | | trajectory to image | | | | |
|---|---|---|---|---|---|---|---|---|---|---|
| | mAP↑ | R@1↑ | R@5↑ | R@10↑ | medR↓ | mAP↑ | R@1↑ | R@5↑ | R@10↑ | medR↓ |
| CImTr | **8.12** | **4.14** | **10.85** | **15.22** | **178** | **8.44** | **4.37** | **11.37** | **15.58** | **170.5** |
| w/o augmentation | 6.52 | 3.55 | 8.21 | 11.94 | 260.5 | 6.55 | 3.47 | 8.28 | 11.88 | 261 |
| w/o downsample | 5.29 | 2.03 | 7.02 | 10.87 | 193 | 5.34 | 2.08 | 7.13 | 11.35 | 194.5 |
| w/o t2t recons | 7.67 | 4.10 | 10.03 | 14.01 | 217 | 7.81 | 4.35 | 10.19 | 14.06 | 218 |
| w/ VAE | 7.31 | 3.46 | 9.95 | 14.20 | 194 | 7.50 | 3.68 | 10.22 | 14.69 | 193 |
| w/ i2t recons | 7.39 | 3.80 | 9.48 | 13.69 | 199 | 7.57 | 4.00 | 9.92 | 13.75 | 194 |

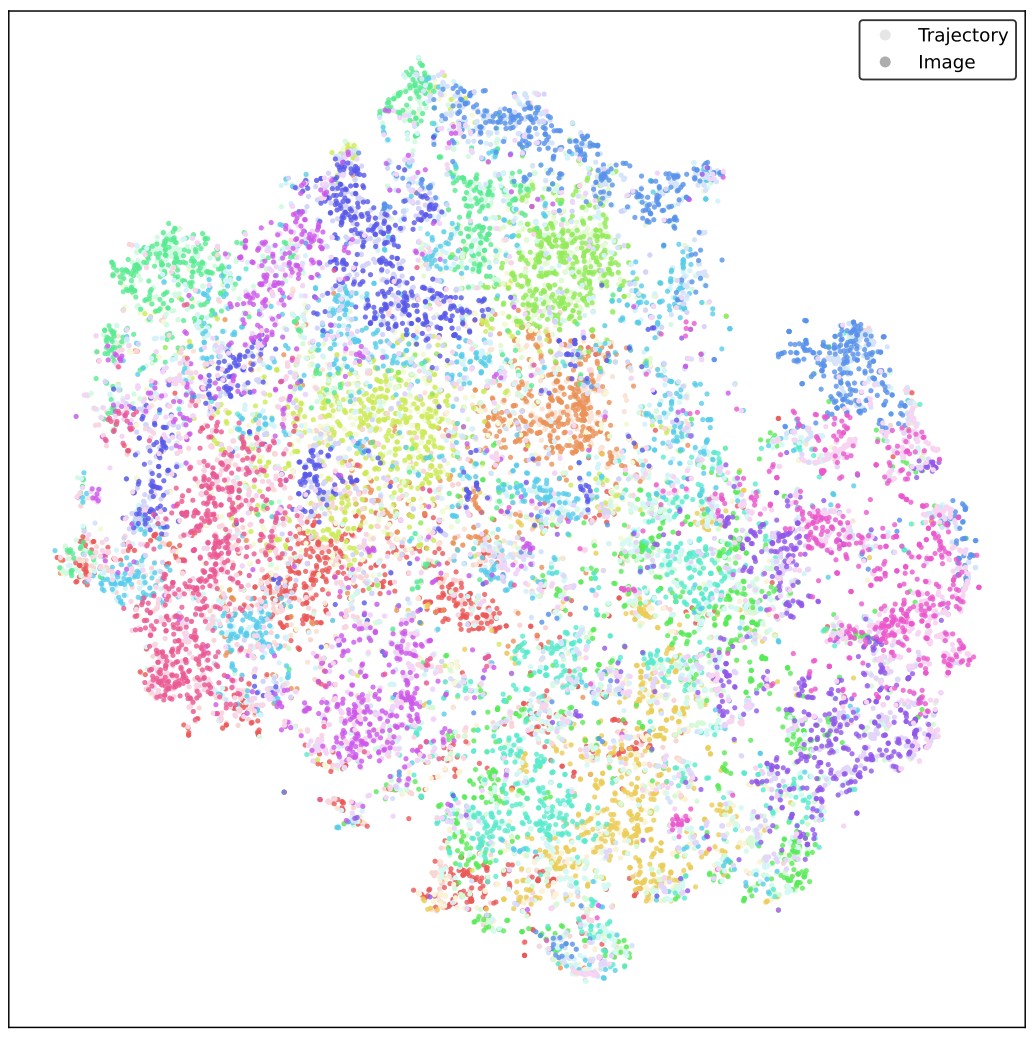

Figure 16: TSNE visualization of the CImTr feature space on DroneMotion-99k validation set.

Table 5: Details of the modules in DVGFormer.

| function | specification |
|---|---|
| camera pose tokenization | 3 MLP layers, 384 dimensions |
| camera motion tokenization | 3 MLP layers, 384 dimensions |
| RGB feature extractor | DinoV2 (Oquab et al., 2023) (fixed) |
| RGB feature projection | AvgPool(5, 9), 2 MLP layers, 384 dimensions |
| depth estimation | DepthAnythingV2 (Yang et al., 2024) (fixed) |
| depth feature | AvgPool(5, 9), 3 convolutional layers, 3x3 kernel, 384 dimensions |
| `<Style>` token | $2 \times 384$ vectors, one for FPV and one for non-FPV drones |
| `<Cond>` token | $1 \times 384$ random vector |
| `<Pred>` token | $1 \times 384$ vector |
| frame-level PE | $30 \times 192$ matrix (10 seconds at 3 fps for images) |
| sub-frame-level PE | up to $52 \times 192$ matrix (1 pose, 45 image patches, 1 `<Pred>`, 5 motions per image) |
| auto-regressive transformer | GPT-2 architecture, 12 layers, 6 heads, 384 dimensions |

latent dimension of 256. Specifically, we have a $5\times$ downsample rate for the trajectory projector and unprojector network (upsample), shrinking the temporal sequence from 15 fps raw motion data to 3 fps. For the image projector, we downsample the feature map to a resolution of $5 \times 9$ before two MLP projection layers.

We use two RTX 3090 to train the CImTr metrics with a batch size of 512 on each GPU and a learning rate of $1 \times 10^{-3}$ for 200 epochs on the training set of DroneMotion-99k.

During training, we incorporate the same data augmentation strategy as DVGFormer, including temporal clipping, horizontal flipping, image scaling, and color jittering. Specifically, for horizontal flipping, we apply this augmentation to not only the image streams but also the camera trajectory, so as to maintain multi-view consistency (Hou & Zheng, 2021).

We report the retrieval performance of the proposed CImTr feature on the validation set of the DroneMotion-99k dataset in Table 4. Over 10141 validation clips, the CImTr feature can achieve a mean average precision of 8.12% and 8.44% on the image-to-trajectory (i2t) and trajectory-to-image (t2i) settings, respectively, which is the best across all variants in Table 4. Similarly, for top-1, top-5, and top-10 retrieval accuracy (R@1, R@5, R@10), and the median rank (medR), CImTr also achieves the best result.

We also point out that although the raw mAP values are low (around 8%), this mainly reflects the challenge in the shot planning task, and the retrieval task itself (10141 samples and only one true match).

We also verify the effectiveness of the proposed feature representation. We find the data augmentation, the projector network for temporal and image downsampling, and the trajectory-to-trajectory (t2t) reconstruction loss all essential to the CImTr design. On the other hand, the variational auto-encoder (VAE) design (Kingma et al., 2019) and the image-to-trajectory (i2t) reconstruction loss, two design highlights in TMR (Petrovich et al., 2023) and CLaTr (Courant et al., 2024), are found not useful for improving the retrieval performance.

We finally visualize the learned embedding space of CImTr in Fig. 16. We first cluster the image features using K-means. Next, we color the image and trajectory features correspondingly in the TSNE plot. We have a few findings:

- The image features are very diverse and do not exhibit strong cluster-like distributions. This reflects the difficulties in the unguided setting, where the task cannot be easily classified into clusters first.

- The trajectory features are also very diverse and are difficult to cluster, suggesting the challenges in the guided setting.

- The CImTr feature can bring image and trajectory features closely together, verifying the effectiveness of our design.

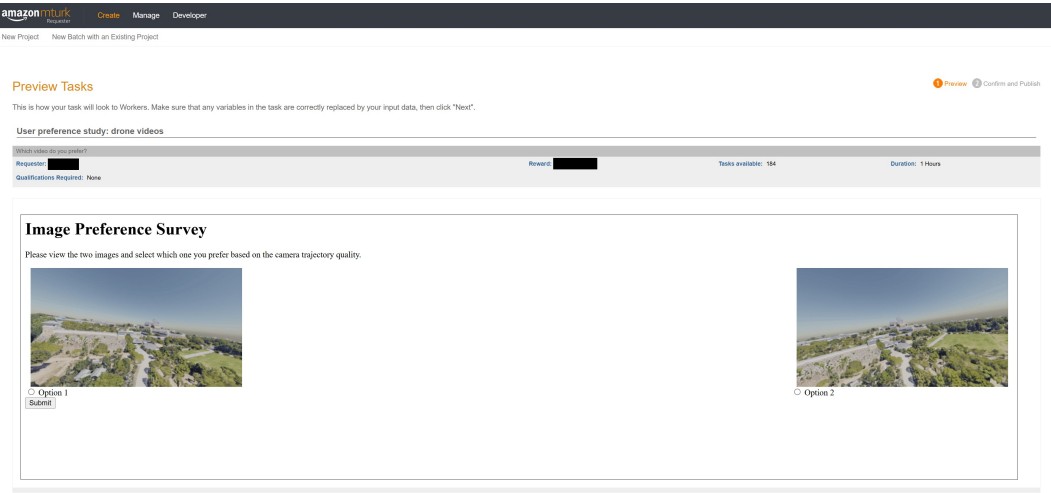

Figure 17: Human preference study using GIFs.

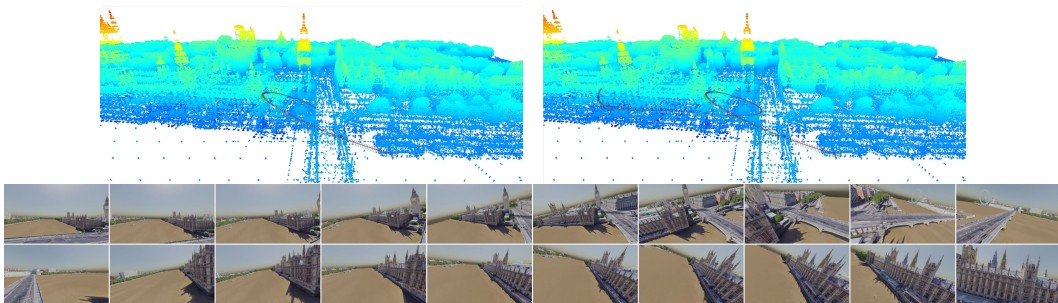

Figure 18: DVGFormer can extend 10-second-long sequence (**top left** and **first row** of the frames) to generate 20-second-long videos (**top right** and **first row** of the frames).

## D    DVGFORMER DETAILS

For camera pose and motion tokenization, we adopt 3 MLP layers with 384 hidden dimensions. For DinoV2 (Oquab et al., 2023) feature projection, given image resolution of $168 \times 294$, we first downsample the feature map from $12 \times 21$ to $5 \times 9$ with average pooling. Then, we apply two MLP layers with 384 hidden dimensions. For the monocular depth estimation results from DepthAnything (Yang et al., 2024), we also average pool the depth map from $168 \times 294$ to $5 \times 9$, and apply three convolutional layers with $3 \times 3$ kernel size and 384 hidden dimensions. We use GELU activation (Hendrycks & Gimpel, 2016) for all modules unless specified. For positional embeddings (PE), we consider 192 dimensions for both frame-level PE and sub-frame-level PE, and concatenate them together into an overall PE of 384 dimension before adding to the tokens. We list all modules in DVGFormer in Table 5.

## E    OTHER EXPERIMENTATION DETAILS

We do not use the image generation metrics like PSNR or text-to-video evaluation metrics in VBench(Huang et al., 2024). In terms of 2D image, since we do not modify 2D pixel, the 2D image PSNR would only reflect the rendering quality in Blender, which itself is adjustable based on the computational budget. Same goes for the image-based metrics of Appearance Style, Scene, Color, Multiple Objects, Object Class, and Imaging Quality in VBench. As for temporal evaluation for videos, VBench metrics including Motion Smoothness, Temporal Flickering, Background Consistency, Subject Consistency, Overall Consistency, are not applicable either, since the temporal

consistency between neighboring frames are also guaranteed by the video rendering pipeline in Blender.

Based on the spec sheet from DJI Avata 2 (FPV drone) and DJI Mavic 3 Pro (non-FPV drone), we set the camera focal lengths for the two types of drones to 12 mm and 24 mm. We do not restrict the angular velocity of either drone types and rely on DVGFormer to predict the angular motion. For linear speed, due to the scale ambiguity, we set the typical speed to 15 m/s for FPV drones and 7.5 m/s for non-FPV drones.

**User study details**. We conduct human evaluation using the AWS Mechanical Turk platform and host online questions with side-by-side visual comparisons of the videos. A screenshot of the online human evaluation preview is shown in Fig. 17. We hired five participants to compare two DVGFormer models (DVGFormer-A and DVGFormer-C) and two baselines (DIRECTOR (Courant et al., 2024) and GenDoP (Zhang et al., 2025)) against the reference model DVGFormer-B. For each of the four comparisons, we present 184 GIFs with the same initial camera pose side-by-side. We provide compensation of 0.04 USD per video pair. The potential risk of photosensitive seizures has been fully disclosed to the participants before accepting the job. The user preference study was internally approved.

**Extending existing camera path**. We show an example of predicted camera paths for different video durations in Fig. 18. The original 10-second-long camera path is shown on the top left subfigure and the first row of the frames. When given a longer duration, the proposed model can extend the existing trajectory in a very smooth manner, into the full 20-second-long sequence on the top right and the bottom row of the frames.

