# OpenReview forum: "Learning to Fly Camera Drones by Watching Internet Videos"
_ICLR.cc/2026/Conference — ICLR 2026 Conference Withdrawn Submission_

### Official Review · Reviewer_ieSR · 2025-10-28

**Soundness:** 2
**Presentation:** 2
**Contribution:** 2
**Rating:** 2
**Confidence:** 4

**Summary:**

This paper addresses the challenge of automating camera drone videography, where it is difficult to define reward functions that capture human aesthetic preferences.

Instead of relying on reinforcement learning, the authors propose an imitation learning framework that learns from Internet drone videos. The authors used COLMAP to reconstruct 3D camera trajectories from online footage and introduced a UKF-based filtering scheme to ensure motion smoothness and trajectory quality. After filtering out noisy reconstructions, they compile the DroneMotion-99k dataset, containing 99k high-quality 3D trajectories (≈180 h total), making it the largest of its kind.

To evaluate this new task, the authors design an interactive simulation environment in Blender with 38 synthetic and 7 real-city scenes, where drone trajectories are rendered and assessed under interactive control. They propose new evaluation metrics at both the instance and dataset levels: (i) kinetic features to quantify motion dynamics, (ii) CImTr (Contrastive Image-Trajectory) features to measure image–trajectory alignment, along with standard indicators such as crash rate, diversity, and human preference.

Finally, they introduce DVGFormer, a transformer-based baseline that predicts camera motions conditioned on past frames, poses, and a style token (FPV vs non-FPV). Three variants (A/B/C) explore different conditioning and prediction strategies. Experiments show that interactive motion prediction (B/C) outperforms offline baselines and existing models, achieving smoother, more diverse, and image-aligned trajectories.

**Strengths:**

- Novel data source and task formulation: First large-scale dataset for drone videography motion imitation directly from online videos.

- Careful data curation: The UKF-based smoothness filter significantly improves reconstruction quality.

- New benchmark setup: Interactive simulation and well-designed quantitative metrics tailored to camera motion.

- Comprehensive analysis: Includes ablations, user preference studies, and generalization to unseen scenes.

**Weaknesses:**

- One of my largest concerns is despite aerial cinematography being a very practical and relevant task, I am not certain if/how the setting the authors proposed here can be relevant. One of the significant benefits of using a drone for videography is that it can perform various styles/maneuvers depending on different requirements. Here, I think the work overclaims the "style" reference in the paper, as it is essentially a binary variable indicating whether the drone is an FPV drone or not. Hence, I doubt how the formulation the authors proposed can be practically used.
- Although the authors formulate the problem as a strong end-to-end computer vision task, I am wondering how a simple classical baseline using Visual Odometry (with scale ambiguity, similar to the authors' setting as well) can perform the task by using the state to simply track a predefined trajectory up to a scale. This experiment could better justify the usage of massive pretraining for this task.
- Following up on insufficient problem motivation, I feel the paper lacks strong novelty or important insights. The major contributions include utilizing internet datasets and training an action regression model, as well as benchmarking against several methods.
- Sim2real consideration: From my understanding, the paper mainly uses the synthetic dataset (including rendered "real city scans"). How could the benchmark potentially guide the decision choices to be made based on real images? Some evaluations/benchmarkings on the real video footage would be very helpful and make the results more convincing. Since in the real world, FPV video can contain many quality degradations, such as motion blur.
- It is nice to mention the implementation on the drone hardware as a good future direction, but another important thing to consider for this is how the current model is capable of running onboard. It would be crucial to report how fast the current model, DVG Former (which takes many frames and action sequences as input), can run on some embedded platforms (e.g., a Jetson Orin). This should also be a very interesting and insightful metric for the benchmark as well.

**Questions:**

See weaknesses above.

---

### Official Review · Reviewer_494u · 2025-10-30

**Soundness:** 3
**Presentation:** 3
**Contribution:** 2
**Rating:** 4
**Confidence:** 5

**Summary:**

Since Videography lacks a clean reward, the paper formulates drone camera control as imitation learning from Internet videos, reconstructing 3D camera poses and learning to imitate expert motions.

They collected  trajectories  from YouTube by SfM (COLMAP) plus Kalman-based smoothness filtering, which is  claimed as the largest camera-motion set of such kind.
Then an interactive Blender testbed renders frames conditioned on actions, across 38 synthetic natural scenes and 7 real city scans, with metrics for crash rate, human preference, image–trajectory alignment, and distributional quality/diversity (FID on kinetic/CImTr features).
DVGFormer (autoregressive Transformer) predicts camera motion/pose from recent frames and tokens, showing behaviors like orbiting, reveals, and obstacle-aware weaving.

Overall, this is an interesting application but contains no novelty in terms of learning and simulation-only results. Due to monocular scale issues, both training and metrics use normalized scale, absolute speed/size realism cannot be established. So it is doubtful whether the actual drone flight in this manner can be performed in real world.

**Strengths:**

Learning from videos sidesteps the missing reward in cinematography and yields a large, inexpensive dataset.
Automatic UKF smoothing filter with a labeled ROC-based threshold to reject jittery reconstructions; yields usable trajectories.

**Weaknesses:**

Due to its nature, monocular SfM scale is normalized, and here real-world absolute speeds are injected heuristically. This limits external validity.
Since this is RGB-only perception by default,  3D perception remains an open problem.
Metric learning signal is weak. The learned alignment metric might be useful but coarse as a proxy for good shots.
They provide simulation only (with low fidelity), and it does not look transferable to actual hardware tests due to the above-mentioned fundamental limitations as well as the computation speed if used onboard a small drone.
Also, authors acknowledge crash rate is still too high for real drones and that low-level motor control is out of scope.

**Questions:**

How can this result turn into real-world implementation, in terms of computation speed (and control requirement for drones), fundamental limitation of monocular visual navigation, and sim-to real issues including varying scenes and environments ?

With scale normalized and low-res renders, how predictive are the metrics (FID-kinetic, CImTr-S, crash) of real-drone safety and viewer preference?

How does the Kalman-filter trade off recall vs. precision in keeping valid trajectories?


Which ingredients (image context length, motion-vs-pose target, bi-level PE, ...) most affect crash vs aesthetics trade-offs across scenes?

---

### Official Review · Reviewer_HhTN · 2025-10-31

**Soundness:** 3
**Presentation:** 3
**Contribution:** 3
**Rating:** 6
**Confidence:** 4

**Summary:**

This paper proposes an imitation learning framework for teaching camera drones to perform cinematic videography by learning from Internet videos rather than human teleoperation or reinforcement learning with handcrafted rewards. The key contributions are:
1. A massive dataset of 99k high-quality 3D drone trajectories extracted from YouTube videos.
2. A Blender-based simulator with 38 synthetic natural scenes and 7 real-world city scans, plus new metrics for evaluating trajectory smoothness, crash rate, diversity, and human preference.
3. A Transformer-based imitation model that predicts drone camera trajectories conditioned on prior frames and motions, demonstrating behaviors like orbiting, scenic reveals, and obstacle-aware motion.
4. Quantitative metrics such as CImTr-S (image-trajectory alignment), FID on kinetic and trajectory features, and a human preference study to assess video realism and smoothness.

**Strengths:**

1. The paper tackles drone videography, a domain distinct from navigation or racing emphasizing aesthetic motion control. It shifts focus from reward-based reinforcement learning to data-driven imitation learning from large-scale Internet videos.
2. The DroneMotion-99k dataset is an original, large, and high-quality contribution, filling a critical gap for 3D camera trajectory data. Detailed pipeline from video scraping, 3D reconstruction (COLMAP), Kalman filtering, to dataset curation.
3. The interactive evaluation environment adds rigor beyond static benchmarks.

**Weaknesses:**

Dataset limitions:
1. The filtering criterion based on smoothness could bias trajectories toward overly stable motions, potentially removing complex yet valid cinematographic behaviors.
2. Dataset diversity, while large in scale, is still dominated by YouTube’s stylistic biases, possibly affecting generalization to other contexts.
Model limitations:
1. DVGFormer, while effective, is architecturally simple — primarily an adaptation of GPT-like autoregressive models — offering limited novelty on the modeling side.
2. No ablation on dataset size vs. performance, which would clarify how much Internet-scale data contributes to learning quality.

**Questions:**

1. Since CImTr-S and FIDkinetic are new, how well do they correlate with human ratings across diverse scenarios? Could the authors report correlation coefficients or inter-rater agreement scores?
2. Could reinforcement learning fine-tuning on top of imitation improve crash avoidance?
3. Any experiments that can demonstrate how much Internet-scale data contributes to learning quality?

---

### Note · Authors · 2025-12-01

I have read and agree with the venue's withdrawal policy on behalf of myself and my co-authors.